# Fracture Evolution between Blasting Roof Cutting Holes in a Mining Stress Environment

**Min Tu [1], Gaoming Zhao [1,\*], Xiangyang Zhang [1], Qingwei Bu [1,2] and Jiaxin Dang [1]**

[1] Key Laboratory of Safety and High-Efficiency Coal Mining of Ministry of Education,
School of Mining Engineering, Anhui University of Science and Technology, Huainan 232001, China;
mtu@aust.edu.cn (M.T.); xyzhang@aust.edu.cn (X.Z.); buqw1988@imust.edu.cn (Q.B.);
dangjiaxin3@163.com (J.D.)

[2] School of Mining and Coal, Inner Mongolia University of Science and Technology, Baotou 014010, China

\* Correspondence: 2020200109@aust.edu.cn; Tel.: +86-150-8479-6704

**Abstract:** The problem of strong ground pressure caused by thick and hard roofs in coal mine stopes has become more prominent. Blasting roof cutting and pressure relief are effective technical ways to solve this engineering problem. The key to achieving roof cutting and pressure relief is the evolution and penetration of cracks between blasting roof cutting holes in a mining stress environment. To solve this problem, research on the evolution of cracks between blasting roof cutting holes in a mining stress environment is being conducted. Along with specific engineering examples, the mechanical effect of blasting dynamic load in the surrounding rock medium between holes in a mining stress environment is analyzed using the research methods of field investigation, blasting simulation, and mechanical analysis. The fracture evolution law between blasting roof cutting holes under the conditions of different hole diameters and spacings, confining pressure environments, and rock mass strengths is analyzed using the univariate comparative analysis method. The research shows that the superposition effect of blasting dynamic load between adjacent blastholes in the transmission process leads to the "X" type penetration evolution of surrounding rock fracture area and fracture area between blastholes. Among them, the blasting effect of a hole diameter of 70 mm and a hole spacing of 0.8 m is best. When designing the blasting roof cutting scheme, we should pay attention to the key factors such as hole diameter and hole spacing. The hole spacing is particularly critical to the effect of roof cutting and crack formation, whereas the confining pressure and rock mass strength have little impact on the expansion and development of blasting cracks, which can be listed as secondary factors for comprehensive analysis in design. According to the transmission law of blasting dynamic load between holes, the critical criterion of blasting top clearance penetration under mining stress environment is deduced. Four important factors affecting blasting top cutting effect are simulated and studied for optimizing the design of roof cutting scheme. Practical engineering application results show that the blasting roof cutting scheme achieves a good seam-forming effect and creates good initial conditions for thick and hard roof cutting. It can serve as a reference for the decision making of blasting top caving technology under similar engineering conditions.

**Keywords:** thick and hard roof; blasting crack evolution; inter-hole crack penetration; roof cutting; seam forming

## 1. Introduction

As coal mining depth increases, geological conditions such as thick and hard roofs are becoming more common. The lateral and rear overhanging range of the working face, which forms during the mining process, is greater under thick and hard roof conditions. Strong mining pressure appears, which has a severe impact on the stability of roadway bearing capacity. According to relevant studies, using pre-splitting blasting roof cutting technology to achieve roof directional splitting reduces the lateral cantilever length of the

roof and the mining pressure on the roadway and has achieved good application effects in many mines [1,2]. At present, blasting roof cutting has become one of the effective technical ways to solve this engineering problem. The key to achieving the goal of roof cutting is to evolve and penetrate cracks between blasting roof cutting holes. As a result, thick and hard roofs can be cut off along the crack direction, reducing the suspended roof size.

In recent years, many experts and scholars have conducted considerable research on the pre-splitting blasting of thick and hard rock formations. Wang et al. [3] used deep-hole pre-splitting blasting technology to control the roof cutting and collapse of a shallow coal seam face in the Shendong mining area to avoid or reduce large-scale roof pressure. Jun [4] analyzed the explosion stress wave propagation and rock-breaking effect of different explosives. Liu et al. [5] used the blasting simulation test system built in the laboratory to study the crack propagation and mechanical characteristics generated by explosive blasting in different coal and rock media. Wang et al. [6] used numerical simulation to study the crack expansion range of deep blasting. Lv et al. [7] used numerical simulation to study the expansion law of coal-seam, deep-hole-shaped energy-blasting fractures. Zhao et al. [8] studied the effective fracturing range of coal seam deep-hole energy-accumulating blasting via mathematical model calculations. Dai [9] comprehensively considered the strain rate effect of rock under the impact load and the stress state of the actual rock, and based on the Mises strength criterion, derived the calculation formula for the radius of the fracture zone and the fracture zone formed by the explosion of the columnar explosive in the rock. Far and Wang [10] posited a probability prediction formula of the radius of blasting fracture zone through research. In addition, many researchers have used numerical simulation software, such as FLAC, AUTODYN, and LS-DYNA to explore the impact of explosive impact loads on crack propagation [11–13]. Cai et al. [14] used LS-DYNA to simulate the process of coal blasting and discussed the propagation characteristics of the explosion stress waves and expansion of coal cracks during blasting. Yang et al. [15] used LS-DYNA to discuss the effect of the explosion stress waves and explosion gas on the blasting medium during the blasting process. Zhou et al. [16] used LS-DYNA simulation to study the propagation law of the explosion stress waves with different hole spacings.

Many of the abovementioned documents are based on the research on the evolution of cracks in a single-hole blasting in a laboratory environment. However, there is limited research on the evolution of cracks between blasting top holes in a mining stress environment. Sensitive factors affecting the evolution of cracks between blasting roof cutting holes under mining stress and related issues, such as the critical penetration criterion, need to be further investigated. Therefore, this article combines specific engineering examples, adopts field investigation, simulation, mechanical analysis, and other research methods, analyzes the appearance of strong mining pressure caused by the hard roof of a mining site, and examines dynamic loading of the blasting under mining stress environment. In this study, we analyze the mechanical propagation characteristics of the surrounding rock medium of a hole, reveal the evolution characteristics of cracks under the dynamic load of the surrounding rock blasting, and use the univariate comparative analysis method to cut the top of blasting under the conditions of different apertures, hole spacings, confining pressures, and rock mass strength. The evolution law of cracks between holes is analyzed, and the critical penetration criterion of cracks between blasting roof cutting holes in a mining stress environment is further derived using the mechanical analysis method, which serves as a reference for blasting and topping technology decision making under similar engineering conditions.

## 2. Engineering Background Analysis

The main mining 3–1 coal in the 113103 working face of Bojianghaizi coal mine in Ordos has an average buried depth of 670 m, an average mining thickness of 4.64 m, and an average inclination of 3°. The roof is managed by the total caving method. The return airway of the working face is 5.2 m wide and 3.8 m high; the width of the coal pillar is 11 m,

and the anchor beam provides net support. The columnar distribution of the structure of roof rock strata is shown in Figure 1.

| Rock | Columnar | thickness |
|---|---|---|
| Medium sandstone | | 16.2m |
| Conglomerate | | 60.1m |
| Sandy mudstone | | 1.51m |
| Fine sandstone | | 15.1m |
| Mudstone | | 2.59m |
| 3-1 Coal | | 4.64m |
| Mudstone | | 1.04m |
| Fine sandstone | | 3.11m |
| Coal line | | 0.8m |
| Mudstone | | 1.08m |

**Figure 1.** Column chart of roof structure.

Field investigations have revealed that the roof is composed of mudstone- and fine sandstone-composite thick and hard roof rock formations, and the length of the suspended roof on the goaf side is large, causing the return airway to be severely affected by mining. The floor heave of the roadway reaches 1.5–2.0 m, and the roof is anchored. The de-anchoring phenomenon occurs, and the roadway section shrinkage rate near the exit of the working face is more than 50%, as shown in Figure 2. The roadway section is reduced, wind speed is high, with dust flying, and movement of pedestrians and transportation of materials are difficult, which severely affects the production safety of the working face. Therefore, given the above problems, during the mining period, the roof of the roadway is blasted to cut the top and relieve the pressure to ensure safe mining.

Affected by the mining of the previous working face, the top slab of this working face broke into a cantilever beam structure, where Block A was bent and deformed under the action of the overlying rock load. Due to the long cantilever, the mining roadway was under the mining pressure environment. Block A is broken by blasting and topping, reducing its cantilever length and the rotational force acting on the roadway. The key technology when blasting the roof is that each blast hole crack can expand along the line of the blast hole and form inter-hole cracks that evolve and penetrate, causing the entire Block A to be cut off along the strike.

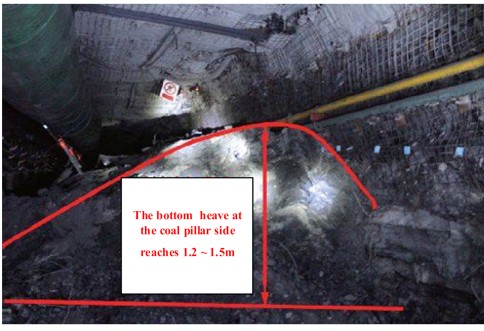

(a)

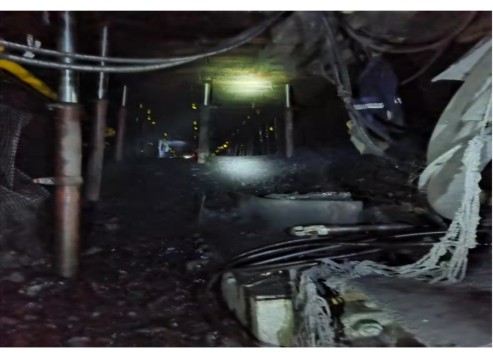

(b)

**Figure 2.** Roadway deformation diagram:(**a**) Floor heave, (**b**) Smaller roadway section.

## 3. Analysis of the Evolution Characteristics of Cracks between Blasting Roof Cutting Holes

Aiming at the key issue of crack penetration between blasting roof cutting holes, this section investigates the evolution characteristics of cracks between blasting roof cutting holes.

### 3.1. Evolution Characteristics of Surrounding Rock Fissures in Single-Hole Blasting

Taking the cut-roof blasting of the return air tunnel in the 113103 working face as an engineering background, a single-hole rock explosive blasting model was developed using ANSYS/LS-DYNA. The initial conditions of the model are as follows: the diameter of the blast hole is 7 cm, confining pressure of the model is 20 MPa, and the rock mass material parameters and explosive state equation are shown in Tables 1 and 2. We established a numerical model consisting of explosives, rocks, and air, and adopted a fluid–solid coupling algorithm. At the same time, to simulate the blasting process in an infinite rock and eliminate the influence of the reflection and superposition of stress at the boundary of the developed model on crack propagation, the surrounding borders were all set as nonreflective borders [17].

**Table 1.** Explosive and its equation of state parameters.

| Density/(g · cm$^{-3}$) | Detonation Velocity/(m s$^{-1}$) | Burst Pressure/(GPa) | $A_1$ | $B_1$ | $R_1$ | $R_2$ | $\omega$ | $E_0$/(GPa) |
|---|---|---|---|---|---|---|---|---|
| 1600 | 6900 | 4.0 | 214.4 | 0.182 | 0.9 | 0.15 | 0.15 | 4.192 |

$A_1$: = Equation of state coefficient, $A_1$.; $B_1$: = Equation of state coefficient, $B_1$.; $R_1$: = Equation of state coefficient, $R_1$.; $R_2$: = Equation of state coefficient, $R_2$.; OMEG: = Equation of state coefficient, $w$.

**Table 2.** Rock mass material parameters.

| Density/ (g · cm⁻³) | Elastic Modulus/(MPa) | Poisson's Ratio $\mu$ | Yield Stress/(MPa) | Tangent Modulus/(GPa) | Hardening Coefficient | Dynamic Tensile Strength/(MPa) |
|---|---|---|---|---|---|---|
| 2700 | 47.7 | 0.37 | 75 | 17.4 | 1 | 3 |

The simulation effect of single-hole blasting is shown in Figure 3. The average radius of the crush zone is 21 cm and average radius of the crack zone is 103 cm. In addition, the crush zone and the crack zone are symmetrically distributed.

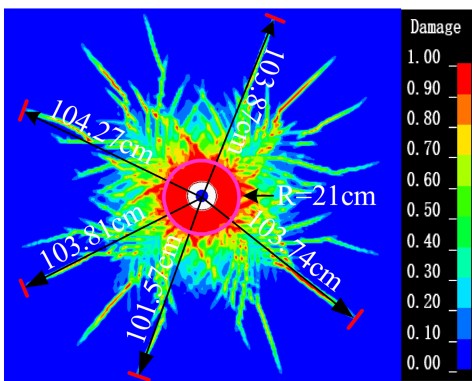

**Figure 3.** Simulated crack propagation diagram of single-hole blasting.

The total energy change curve of single-hole blasting is shown in Figure 4. After the main explosive is detonated, the accumulated detonation energy acts on the surrounding rock of the hole wall to form a high-strength blasting dynamic load and rapidly diffuses from the hole wall to the deep part of the surrounding rock. The detonation energy decays rapidly during the propagation process.

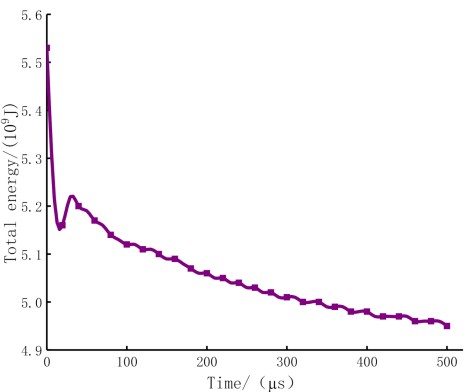

**Figure 4.** Energy variation curve of single-hole blasting.

The dynamic load propagation and radial crack formation process of single-hole blasting are shown in Figure 5. Under the action of the dynamic load of blasting, the blast hole diameter is broken toward the shallow surrounding rock as shown in Figure 5a–c, and the hole diameter forms fissures toward the deep surrounding rock, as shown in Figure 5d–f. At the same time, the explosive detonation energy quickly rushes into the fissures of the surrounding rock and produces a violent dynamic load expansion on the surrounding rock media in the fissures, thereby forming a "divergent" blasting aperture to the fissure distribution characteristics, as shown in Figure 5g–i.

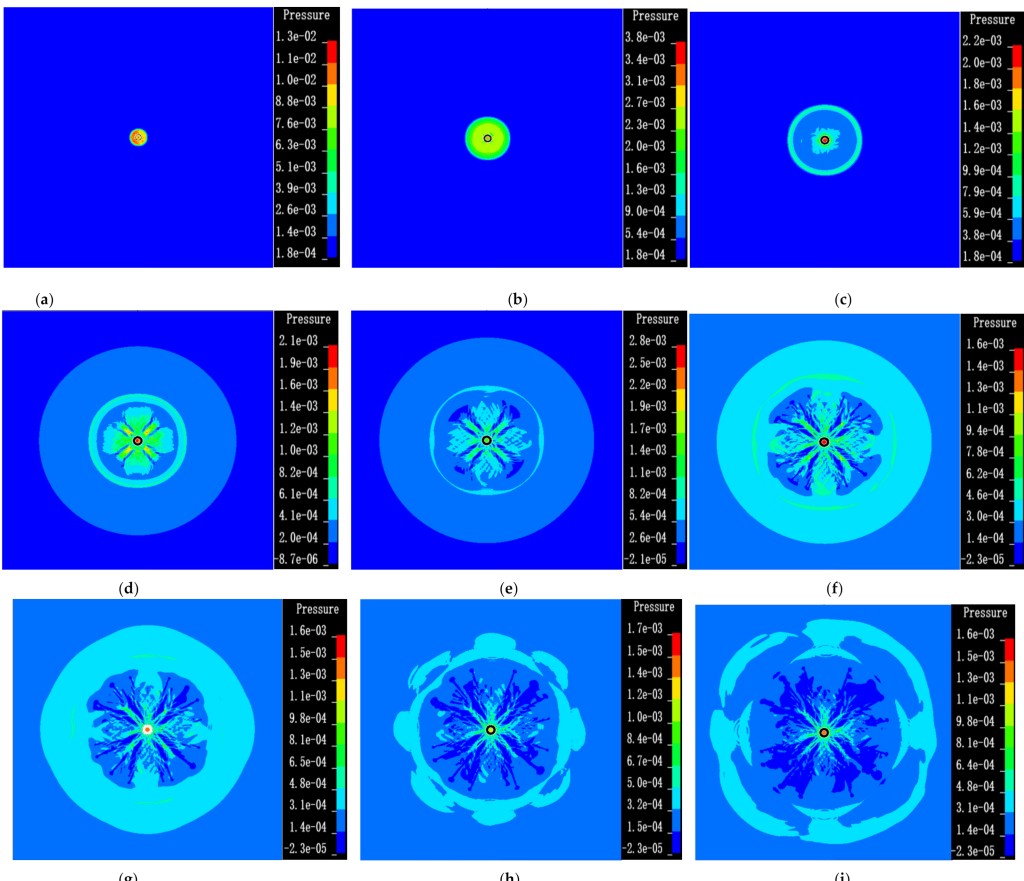

**Figure 5.** Dynamic load propagation process of single-hole blasting. (**a**). Explosion time 29.79 *μs* (**b**). Explosion time 89.77 *μs.* (**c**) Explosion time 159.93 *μs.* (**d**) Explosion time 219.88 *μs.* (**e**) Explosion time 269.84 *μs.* (**f**) Explosion time 359.93 *μs.* (**g**) Explosion time 399.81 *μs.* (**h**) Explosion time 449.95 *μs.* (**i**) Explosion time 500.07 *μs..*

Through the simulation of the radial fissure evolution of single-hole blasting, the following is observed: (1) the dynamic load of blasting is severely attenuated during the diffusion and propagation process, (2) the scale of blasting dynamic load fracturing is limited, (3) detonation energy can easily flow into the fissure space of the surrounding rock and aggravate the expansion of the fissure, and (4) the blast fracture gap extends and evolves from the hole wall to the deep part of the surrounding rock. From this analysis, it is concluded that the key to achieving the goal of blasting cutting and perforating the joint is to make full use of the limited blasting dynamic load, to form an effective inter-hole fissure penetration on the surrounding rock of the hole wall.

### 3.2. The Evolution Law of Cracks between the Top Holes of Blasting

To further reveal the sensitive influencing factors and laws of cut-roof blasting in the mining stress environment, the univariate comparative analysis method is used to analyze the influence of different apertures, hole spacings, confining pressure, surrounding rock strength, and other factors on the evolution of perforation cracks in cut-top blasting. We established a numerical model of blasting top cutting (size: 6000 cm × 3000 cm × 2 cm), as shown in Figure 6. The five holes have no explosives in the middle hole, which is used as a peephole to observe the penetration effect of the burst rupture gap. Confining pressure is uniformly applied to the surrounding nonreflective boundary. The univariate comparative analysis method is used to design the simulation comparison plan, as shown in Table 3.

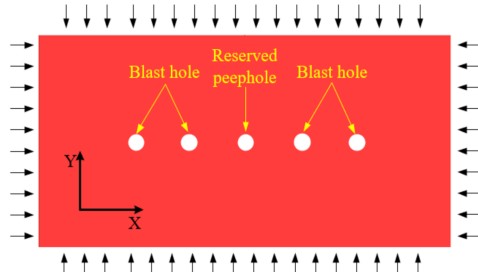

**Figure 6.** Numerical simulation model.

**Table 3.** Simulation scheme of single quantity comparative analysis.

| | Blast Hole Diameter/(mm) | Hole Spacing/(m) | Confining Pressure/(MPa) | Tension–Compression Ratio |
|---|---|---|---|---|
| Option 1 | 30 mm, 50 mm 70 mm, 90 mm | 0.8 m | X, Y: 20 MPa | 0.04 |
| Option 2 | 70 mm | 0.6 m, 0.8 m 1.0 m, 1.2 m | X, Y: 20 MPa | 0.04 |
| Option 3 | 70 mm | 0.8 m | X, Y: 20 MPa 30 MPa 60 MPa, 90 MPa | 0.04 |
| Option 4 | 70 mm | 0.8 m | X, Y: 20 MPa | 0.04 0.0625 0.125 0.25 |

### 3.2.1. Influence of Aperture on the Evolution of Cracks between Roof Cutting Blasting Holes

The simulation results of the blasting rupture gap penetration with different blast hole diameters are shown in Table 4 and Figure 7. The crack between the two blasting holes with a diameter of 30 mm is not penetrated. When the pore size is extended to 50 mm, the number of cracks between two blasting holes is less. With the continuous expansion of the hole diameter, the greater the penetration degree of the reserved peephole crack. However, the larger the hole diameter, the more explosives are required in the blast hole, which increases the cost-effectiveness of the project. Therefore, it is necessary to reasonably determine the parameters of blasting holes in combination with other factors, so as to ensure the connection of cracks between holes and achieve better economic benefits.

**Table 4.** Propagation table of blasting through cracks with different blast hole diameters.

| Blast Hole Diameter/(mm) | Penetration of Reserved Peepholes | Farthest Propagation Distance of Edge Hole Crack/(m) |
|---|---|---|
| 30 mm | Not penetrated | 0.437 m |
| 50 mm | Single fracture penetration | 0.751 m |
| 70 mm | Multi fracture penetration | 1.040 m |
| 90 mm | Crushing penetration | 1.076 m |

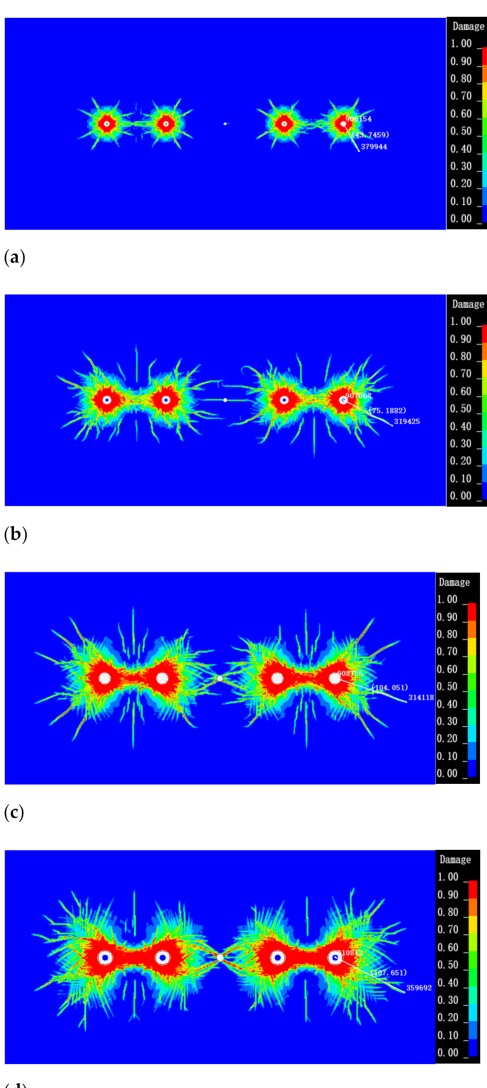

**Figure 7.** Evolution and distribution characteristics of cracks between blasting roof cutting holes with different hole diameters. (**a**) Blasting hole diameter 30 mm. (**b**) Blasting hole diameter 50 mm. (**c**) Blasting hole diameter 70 mm. (**d**) Blasting hole diameter 90 mm.

3.2.2. Influence of Hole Spacing on the Evolution of Blasting Perforation Cracks

As shown in Table 5 and Figure 8, when the hole spacing is 0.6 m and 0.8 m, the peephole and the blasting holes on both sides form a gap between the holes and penetrate closely. When the hole spacing is 1.0 m, only the cracks between the peepholes and the blasting holes on both sides have the potential to approach and penetrate. When the hole spacing is 1.2 m, the peephole and blasting holes on both sides fail to form a crack between the holes to penetrate closely. The blast hole diameter determines the action scale of blasting dynamic load and the cracking range of surrounding rock, and the hole spacing is the effective stroke of crack penetration between holes. In the design scheme of blasting roof cutting, the blast hole diameter and hole spacing of blasting roof cutting holes need to be designed together, and the limited blasting dynamic load should be fully used to achieve the expected goal of cutting the top blasting and piercing into a seam. In order to determine the optimal hole spacing in the engineering site, the damage range of rock mass under different hole spacing is discussed by means of simulation, and then the optimal row spacing between blast holes is determined.

**Table 5.** Propagation of blasting through cracks with different hole spacings.

| Hole Spacing/(m) | Penetration of Reserved Peepholes | Farthest Propagation Distance of Edge Hole Crack/(m) |
|---|---|---|
| 0.6 m | Crushing penetration | 1.075 m |
| 0.8 m | Fissure penetration | 1.040 m |
| 1.0 m | Not penetrated | 0.964 m |
| 1.2 m | Not penetrated | 1.034 m |

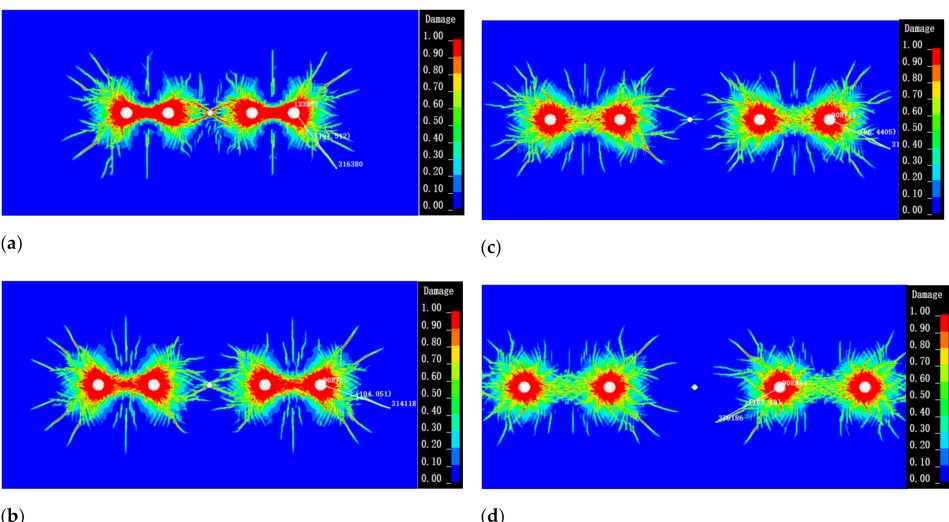

(a)      (c)

(b)      (d)

**Figure 8.** Evolution and distribution characteristics of cracks between blasting roof cutting holes under different hole spacings. (**a**) Blasting hole spacing 0.6 m. (**b**) Blasting hole spacing 0.8 m. (**c**) Blasting hole spacing 1.0 m. (**d**) Blasting hole spacing 1.2 m.

### 3.2.3. Influence of Different Confining Pressures on the Evolution of Blasting Perforation Cracks

As shown in Table 6 and Figure 9, under different confining pressure conditions, the reserved peepholes have crack penetrations. Confining pressures of 20–30 MPa have minimal effect on the blasting effect. When the confining pressure increases to 60 MPa, the upper and lower rock cracks near the midpoint of the two blasting holes decrease. When the confining pressure increases to 90 MPa, the range of the crush zone between the two blast holes becomes smaller, expansion of the cracks around the single-blast hole is suppressed, and the length and number of cracks are notably reduced. The simulation results show that the greater the confining pressure, the greater the impact of the rock mass on the cracking resistance of the blasting dynamic load, resulting in greater energy loss during the transmission of the blasting dynamic load. Moreover, a longer distance crack expansion effect cannot be produced. However, the confining pressure has minimal effect on the perforation effect of cut-roof blasting within a certain range. When the confining pressure increases to 90 MPa, the fracture area of the blast hole is remarkably reduced. Therefore, when blasting the roof cutting, the stress environment of the roadway roof should be grasped in time. In the case of stress concentration areas, timely changes to the blasting roof cutting plan to increase the explosive amount or reduce the hole spacing improves the roof cutting blasting effect.

**Table 6.** Penetration propagation of blasting cracks under different confining pressures.

| Confining Pressure/(MPa) | Penetration of Reserved Peephole | Farthest Propagation Distance of Edge Hole Crack/(m) |
|---|---|---|
| 20 MPa | Fissure penetration | 1.040 m |
| 30 MPa | Fissure penetration | 1.023 m |
| 60 MPa | Fissure penetration | 0.931 m |
| 90 MPa | Fissure penetration | 0.864 m |

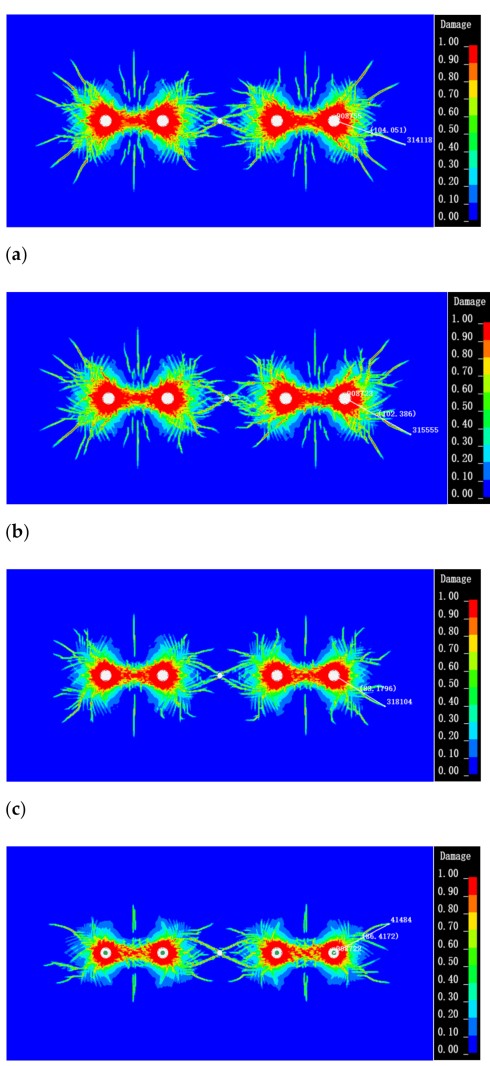

**Figure 9.** Evolution and distribution characteristics of cracks between blasting roof cutting holes under different confining pressures. (**a**) Model applied pressure 20 MPa. (**b**) Model applied pressure 30 MPa. (**c**) Model applied pressure 60 MPa. (**d**) Model applied pressure 90 MPa.

### 3.2.4. Influence of Rock Mass Strength on the Evolution of Blasting Perforation Cracks

As shown in Table 7 and Figure 10, the cracks between the rock blasting holes of different strengths can penetrate the reserved peepholes. The blasting effect is not considerably different when the tension–compression ratio is between 0.04 and 0.125; only the single-hole blasting gap evolution scale is slightly reduced. When the tension–compression ratio increases to 0.25, the expansion range of the single-hole blasting fracture gap decreases rapidly, and the rock fissure near the midpoint of the two blasting holes disappears. The cracks near the blasting hole are arranged in parallel, which is different from the previ-

ous irregular free extension, as shown in Figure 10d. The simulation results show that the strength of the rock mass has minimal influence on the effect of blasting cutting and perforation. However, the expansion of the blast fracture is caused by the micro-cracks in the rock mass under the action of blasting dynamic load. The larger the size, the stronger is the detonation pressure tension it can withstand, resulting in fewer burst rupture gaps and reduced propagation distance. Combined with blasting roof cutting, the rock mass strength determines the difficulty of explosive blasting dynamic load on rock mass breaking and propagation. Designing a reasonable blasting roof cutting scheme under different strengths can effectively increase the crack propagation range and improve the success rate of roof cutting.

**Table 7.** Penetration propagation of blasting cracks with different tension–compression ratios.

| Tension–Compression Ratio | Penetration of Reserved Peepholes | Farthest Propagation Distance of Edge Hole Crack/(m) |
| --- | --- | --- |
| 0.04 | Fissure penetration | 1.040 m |
| 0.0625 | Fissure penetration | 1.036 m |
| 0.125 | Fissure penetration | 1.016 m |
| 0.25 | Fissure penetration | 0.908 m |

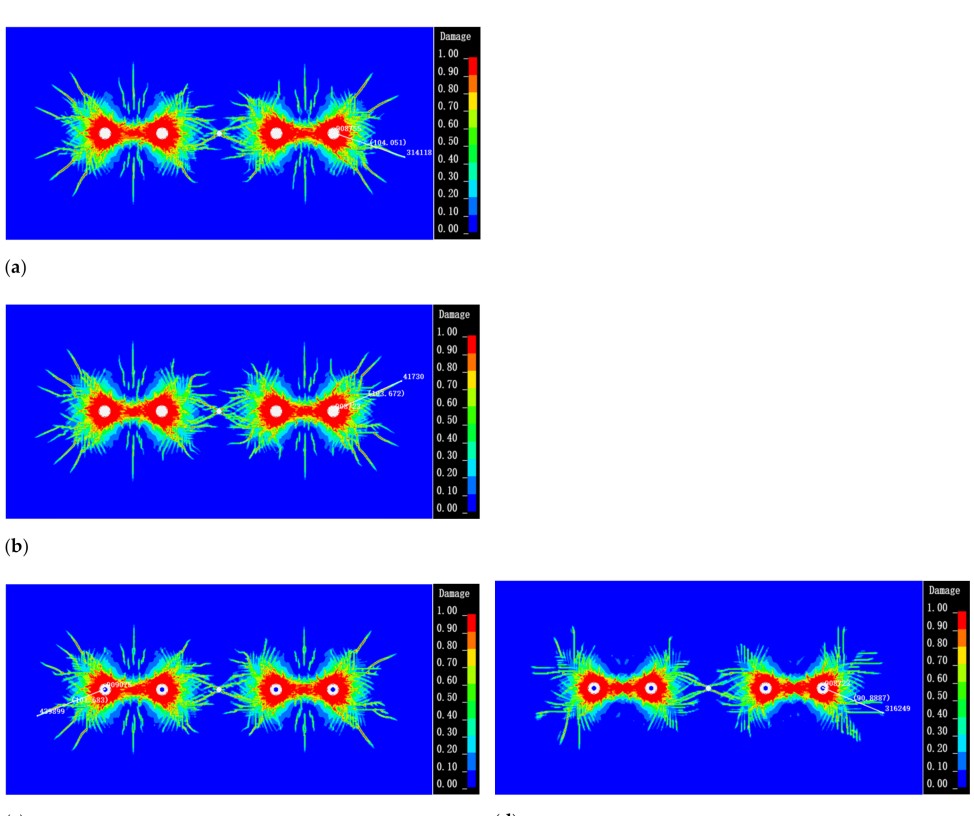

(**a**)

(**b**)

(**c**)                    (**d**)

**Figure 10.** Evolution and distribution characteristics of cracks between blasting roof cutting holes under different tension–pressure ratios. (**a**) Tension compression ratio 0.04. (**b**) Tension compression ratio 0.0625. (**c**) Tension compression ratio 0.125. (**d**) Tension compression ratio 0.25.

## 4. Mechanical Analysis of Crack Penetration between Blasting Cut Holes in Mining Stress Environment

According to the evolution characteristics of cracks between blasting roof cutting holes revealed in the previous section, this section establishes the fracture-forming criterion of blasting roof cutting perforation in a mining stress environment using a mechanical

analysis method and obtains the critical breakdown spacing criterion to assist in optimizing the technical scheme of roof cutting blasting.

After the explosive is detonated, the blasting dynamic load of high temperature and pressure impacts the hole wall on both sides of the hole, and its peak stress is remarkably greater than the compressive strength of the rock mass. The energy of the shock wave decreases rapidly while crushing the rock until the energy is exhausted. Therefore, the explosion impact area can be roughly divided into blasting crushing zone and blasting fissure zone, respectively. The stress and failure distribution of the surrounding rock of the blast hole wall under blasting dynamic load is shown in Figure 11.

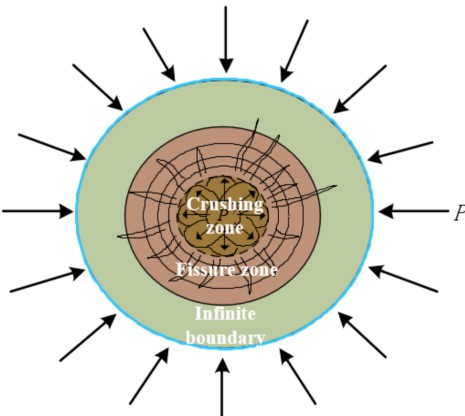

**Figure 11.** Schematic of blasting analysis.

In Figure 11, the inner circle is the range of the crushing area, with a radius of a; the outer circle is infinite, with a radius of m; detonation pressure at the edge of the crushing area transmitted by the dynamic load of explosive blasting is P′ (P represents the size of the external uniformly distributed load); and the radius of the fracture area is set as b.

Under engineering blasting conditions, P1 represents the initial impact pressure on the hole wall [9]:

$$P_1 = n_1 \cdot \frac{\rho_0 V_0}{8} \cdot \left(\frac{d_c}{d_b}\right)^6 \cdot \left(\frac{l_c}{l_b}\right)^2 \tag{1}$$

In the formula, $n_1$ represents the explosion impact pressure coefficient, generally $n_1 = 8$–9; $\rho_0$ represents explosive density; $V_0$ represents explosive detonation velocity; $d_c$ and $d_b$ are roll and blast hole diameters, respectively; and $l_c$ and $l_b$ are the axial charge and axial chamber lengths, respectively.

The radius $r_1$ of the crushing zone of the available cartridge blasting is as follows [9]:

$$r_1 = r_0 \left( \frac{n_1 \left[ \begin{array}{c} (1+\beta)^2 + (1+\beta^2) - \\ 2\mu_d(1-\mu_d)(1-\beta)^2 \end{array} \right]^{1/2} \rho_0 V_0^2}{8\sqrt{2}\sigma_c \sqrt[3]{\varepsilon}} \cdot \left(\frac{d_c}{d_b}\right)^6 \cdot \left(\frac{l_c}{l_b}\right)^3 \right)^{\frac{1}{2+\frac{\mu_d}{1-\mu_d}}} \tag{2}$$

In the formula: $\varepsilon$ is the loading strain rate. In engineering blasting, the rock loading rate $\varepsilon$ is between 10 and 105 s − 1 [18]. In the crushing zone, the loading rate is higher, which can be taken as $\varepsilon$ = 102~104 s − 1; in the fracture zone, the loading rate is further reduced, which can be taken as $\varepsilon$ = 100~103 s − 1, $\beta$ is the lateral stress coefficient, and $\beta = \mu_d/(1-\mu_d)$; $\sigma_c$ is the static compressive strength of the rock mass; $r_0$ is the radius of the blasthole; $\mu$ is the Poisson's ratio of the rock mass, and μd is the dynamic Poisson's ratio. Under engineering blasting conditions, generally $\mu_d = 0.8~\mu$.

Because the blasting effect is affected by numerous factors and the research on this problem is insufficiently comprehensive, the coefficients $\lambda_1$ and $\lambda_2$ are introduced. According to the relevant research [19,20], $\lambda_1 = 0.7$–$0.9$ is used in the crushing zone.

After introducing $\lambda_1$, the radius a of the crushing zone can be obtained as follows:

$$a = \lambda_1 \cdot r_1 \tag{3}$$

According to elastic mechanics, the force at any point in polar coordinates can be expressed as follows [19]:

$$\begin{cases} \sigma_r = \dfrac{\partial^2 \varphi}{\partial y^2} = \dfrac{1}{r}\dfrac{\partial \varphi}{\partial r} + \dfrac{1}{r^2}\dfrac{\partial^2 \varphi}{\partial z^2} \\[2mm] \sigma_\theta = \dfrac{\partial^2 \varphi}{\partial x} = \dfrac{\partial^2 \varphi}{\partial r^2} \\[2mm] \tau_{rz} = -\dfrac{\partial^2 \varphi}{\partial x \partial y} = -\dfrac{\partial}{\partial r}\left(\dfrac{1}{r}\dfrac{\partial \varphi}{\partial z}\right) \end{cases} \tag{4}$$

Under axisymmetric conditions, the Airy stress equation of the surrounding rock of the blast hole wall is:

$$\varphi = \varphi(r) \tag{5}$$

Under axisymmetric conditions, the stress component equations of the surrounding rock of the hole wall are as follows:

$$\begin{cases} \sigma_r = \dfrac{1}{r}\dfrac{\partial \varphi}{\partial r} \\[2mm] \sigma_\theta = \dfrac{\partial^2 \varphi}{\partial r^2} \\[2mm] \tau_{rz} = \tau_{zr} = 0 \end{cases} \tag{6}$$

Therefore, the compatible equation for the problem between axisymmetric holes is:

$$\left(\dfrac{d^2}{dr^2} + \dfrac{1}{r}\dfrac{d}{dr}\right)\varphi = 0 \tag{7}$$

The general solution is:

$$\varphi = A\ln r + Br^2 \ln r + Cr^2 + D \tag{8}$$

In the formula, $A$, $B$, $C$, and $D$ are undetermined coefficients.

Here, Formula (4) can be expressed as follows:

$$\begin{cases} \sigma_r = \dfrac{A}{r^2} + B(1 + 2\ln r) + 2C \\[2mm] \sigma_\theta = -\dfrac{A}{r^2} + B(3 + 2\ln r) + 2C \\[2mm] \tau_{rz} = \tau_{zr} = 0 \end{cases} \tag{9}$$

From the literature [8], the dynamic load P$'$ of blasting at the edge of the crushing zone can be obtained as follows:

$$P' = P_1/\bar{r}^\alpha \tag{10}$$

In the formula, $\bar{r}$ represents the comparison distance, $\bar{r} = r_i/r_0$, $r_i$ represents the distance from any point to the blast hole center, $r_0$ represents the blast hole radius, $\alpha$ represents the stress wave attenuation coefficient, and $\alpha = 2 \pm \mu d/(1 - \mu d)$, take positive in the shock wave action area and negative in the compressive stress wave action area.

Therefore, the boundary conditions are determined as follows:

At boundary $r = a$,

$$(\sigma_r)_{r=a} = -P' \quad \tau = 0 \tag{11}$$

At boundary $r = m$

$$(\sigma_r)_{r=m} = -P \quad \tau = 0 \tag{12}$$

Combined with the displacement condition, B = 0 can be obtained, and A and 2C can be obtained as follows:

$$
\begin{cases}
A = \dfrac{\left(\lambda_1 \cdot r_0 \left(\dfrac{n_1\left[(1+\beta)^2 + (1+\beta^2) - 2\mu_d(1-\mu_d)(1-\beta)^2\right]\rho_0 V_0^2}{8\sqrt{2}\sigma_c \sqrt[3]{\varepsilon}} \cdot \left(\dfrac{d_c}{d_b}\right)^6 \cdot \left(\dfrac{l_c}{l_b}\right)^3\right)^{\frac{1}{2+\frac{\mu_d}{1-\mu_d}}}\right)^2 m^2 (P - P')}{m^2 - \left(\lambda_1 \cdot r_0 \left(\dfrac{n_1\left[(1+\beta)^2 + (1+\beta^2) - 2\mu_d(1-\mu_d)(1-\beta)^2\right]\rho_0 V_0^2}{8\sqrt{2}\sigma_c \sqrt[3]{\varepsilon}} \cdot \left(\dfrac{d_c}{d_b}\right)^6 \cdot \left(\dfrac{l_c}{l_b}\right)^3\right)^{\frac{1}{2+\frac{\mu_d}{1-\mu_d}}}\right)^2} \\[6ex]
2C = \dfrac{P'\left(\lambda_1 \cdot r_0 \left(\dfrac{n_1\left[(1+\beta)^2 + (1+\beta^2) - 2\mu_d(1-\mu_d)(1-\beta)^2\right]\rho_0 V_0^2}{8\sqrt{2}\sigma_c \sqrt[3]{\varepsilon}} \cdot \left(\dfrac{d_c}{d_b}\right)^6 \cdot \left(\dfrac{l_c}{l_b}\right)^3\right)^{\frac{1}{2+\frac{\mu_d}{1-\mu_d}}}\right)^2 - Pm^2}{m^2 - \left(\lambda_1 \cdot r_0 \left(\dfrac{n_1\left[(1+\beta)^2 + (1+\beta^2) - 2\mu_d(1-\mu_d)(1-\beta)^2\right]\rho_0 V_0^2}{8\sqrt{2}\sigma_c \sqrt[3]{\varepsilon}} \cdot \left(\dfrac{d_c}{d_b}\right)^6 \cdot \left(\dfrac{l_c}{l_b}\right)^3\right)^{\frac{1}{2+\frac{\mu_d}{1-\mu_d}}}\right)^2}
\end{cases}
\tag{13}
$$

According to the problem of a circular hole with an infinite boundary, the radial and circumferential stress of the surrounding rock of the blasting hole are as follows:

$$
\begin{cases}
\sigma_r = -P' - \left(1 - \dfrac{1}{r^2}\left(\lambda_1 \cdot r_0 \left(\dfrac{n_1\left[(1+\beta)^2 + (1+\beta^2) - 2\mu_d(1-\mu_d)(1-\beta)^2\right]\rho_0 V_0^2}{8\sqrt{2}\sigma_c \sqrt[3]{\varepsilon}} \cdot \left(\dfrac{d_c}{d_b}\right)^6 \cdot \left(\dfrac{l_c}{l_b}\right)^3\right)^{\frac{1}{2+\frac{\mu_d}{1-\mu_d}}}\right)^2\right) P \\[4ex]
\sigma_\theta = P' - \left(1 + \dfrac{1}{r^2}\left(\lambda_1 \cdot r_0 \left(\dfrac{n_1\left[(1+\beta)^2 + (1+\beta^2) - 2\mu_d(1-\mu_d)(1-\beta)^2\right]\rho_0 V_0^2}{8\sqrt{2}\sigma_c \sqrt[3]{\varepsilon}} \cdot \left(\dfrac{d_c}{d_b}\right)^6 \cdot \left(\dfrac{l_c}{l_b}\right)^3\right)^{\frac{1}{2+\frac{\mu_d}{1-\mu_d}}}\right)^2\right) P
\end{cases}
\tag{14}
$$

By substituting Formula (14) into the Griffith strength theory $k\sigma_{td} = -\dfrac{(\sigma_1 - \sigma_3)^2}{4(\sigma_1 + \sigma_3)}$, the equation of fracture zone radius can be obtained as follows:

$$
k\sigma_{td} = \dfrac{1}{8P}\left(-2P' + \dfrac{2P}{b^2}\left(\lambda_1 \cdot r_0 \left(\dfrac{n_1\left[(1+\beta)^2 + (1+\beta^2) - 2\mu_d(1-\mu_d)(1-\beta)^2\right]\rho_0 V_0^2}{8\sqrt{2}\sigma_c \sqrt[3]{\varepsilon}} \cdot \left(\dfrac{d_c}{d_b}\right)^6 \cdot \left(\dfrac{l_c}{l_b}\right)^3\right)^{\frac{1}{2+\frac{\mu_d}{1-\mu_d}}}\right)^2\right)^2
\tag{15}
$$

In the formula, *k* represents the strength change coefficient of the rock mass under dynamic load, generally taken as 1.3 [21].

Therefore, the critical criterion [d] for penetrating the blasting roof crack under the mining stress environment is:

$$
\begin{cases}
[d] \leq 2nR \\[2ex]
k\sigma_{td} = \dfrac{1}{8P}\left(-2P' + \dfrac{2P}{b^2}\left(\lambda_1 \cdot r_0 \left(\dfrac{n_1\left[(1+\beta)^2 + (1+\beta^2) - 2\mu_d(1-\mu_d)(1-\beta)^2\right]\rho_0 V_0^2}{8\sqrt{2}\sigma_c \sqrt[3]{\varepsilon}} \cdot \left(\dfrac{d_c}{d_b}\right)^6 \cdot \left(\dfrac{l_c}{l_b}\right)^3\right)^{\frac{1}{2+\frac{\mu_d}{1-\mu_d}}}\right)^2\right)^2
\end{cases}
\tag{16}
$$

In the formula, [d] represents the critical criterion for the penetration of the blasting head gap and n is the effective safety factor, which is taken as 0.8 in the text.

## 5. Engineering Practice

According to the site working conditions, Poisson's ratio is expressed as: $\mu = 0.25$, $n = 9$, explosive density $\rho_0 = 1600$ g/cm$^{-3}$, explosion velocity $V_0 = 6900$ m/s$^{-1}$, loading rate $\varepsilon = 10^3$ s$^{-1}$, compressive strength of rock mass $\sigma_c = 68.68$ MPa, cartridge diameter

$d_c$ = 35 mm, blast hole diameter $d_b$ = 70 mm, axial charge length $l_c$ = 1.2 m, axial chamber length $l_b$ = 1.5 m, and blast hole radius $r_0$ = 35 mm $\lambda_1$ for 0.9. The radius of the crushing zone ($a$ = 20.76 cm) can be obtained by substituting Formulas (2) and (3), followed by the explosive explosion pressure $P_1$ = 4000 MPa, confining pressure $P$ = 20 MPa, radius of crushing area $a$ = 20.76 cm, and tensile strength of rock mass $\sigma_{td}$ = 2.75 MPa, $k$ = 1.3. The value of blasting fracture zone b under confining pressure of 20 MPa can be obtained by substituting Formulas (10) and (16). Through calculation, the range of the blasting crack area under 20 MPa is $b$ = 51.58 cm.

Due to the thick hard fine sandstone with strong integrity of roadway roof, along with numerical simulation, theoretical calculation and analysis, and economic factors, the designed roof cutting angle is 75°, drilling diameter is 70 mm, drilling depth is 11–22 m, and the spacing of blasting holes is 0.8 m, as shown in Figure 12.

According to the actual situation of the project, the buried depth of the coal seam is 670 m, and the confining pressure of 17 Mpa is applied. The simulated blasting results are shown in Figure 13. The blasting holes are broken and penetrated and the reserved peephole is penetrated by cracks. The maximum range of cracks in edge holes is 105 cm and the crack penetration effect between blasting top cutting holes is good.

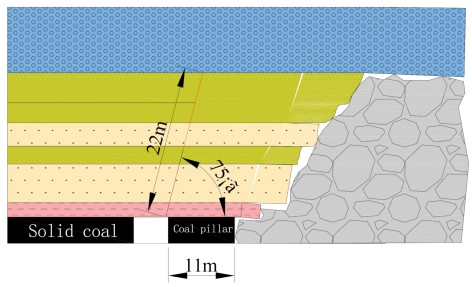

(a)

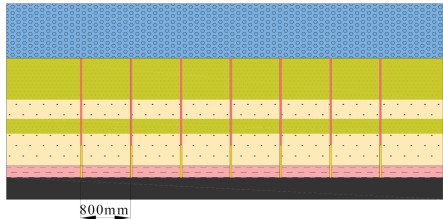

(b)

**Figure 12.** Drilling construction layout. (**a**) Site drilling construction plan. (**b**) Top view of site drilling construction layout.

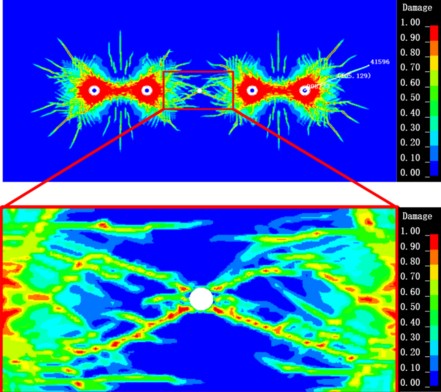

**Figure 13.** Engineering Blasting simulation diagram.

According to the technical scheme of blasting roof cutting, the on-site industrial test is conducted, and the reserved peepholes are selected at different positions for peeping. The on-site peeping results are shown in Figure 14. Obvious cracks can be seen in the peepholes, and the surrounding rocks in some holes collapse locally, indicating that the blasting roof cutting effect is good, and the roof can be completely cut off.

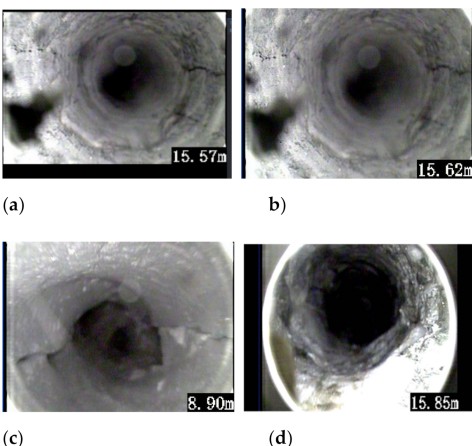

**Figure 14.** Image result diagram of speculum. (**a**) Drilling peep location: back footage 2380 m. (**b**) Drilling peep location: back footage 2387 m. (**c**) Drilling peep location: back footage 2490 m. (**d**) Drilling peep location: back footage 2680 m.

The measured results of roadway surrounding rock deformation before and after roof cutting are shown in Figure 15. More than 180 m in front of the work is affected by advanced mining, of which 60 m in front of the work has the most severe impact. After roof cutting, the average approach of the roof and floor 40 m in front of the working face decreases from 1537 to 1047 mm. The average approach of two sides decreases from 1325 to 837 mm and deformation of roadway surrounding rock evidently decreases. After the roof cutting, the deformation of the roadway 80–200 m behind the adjacent working face gradually tends to be stable, and the integrity of the surrounding rock of the roadway can meet the production and safety requirements. By analyzing the measured results of roadway deformation, it can be seen that blasting roof cutting is conducive to reducing roadway deformation and increasing the stability of roadway and surrounding rock.

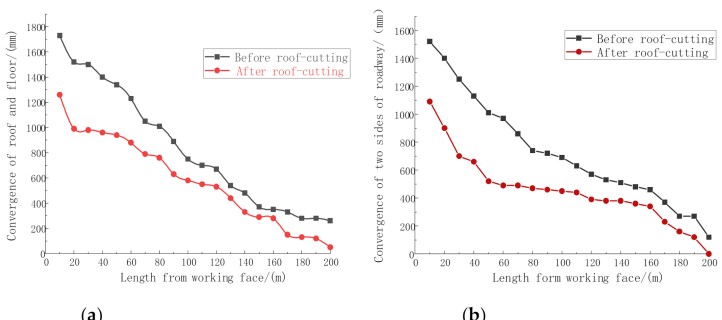

**Figure 15.** Comparison of roadway deformation before and after roof cutting. (**a**) Convergence of roof and floor (**b**) Convergence of two sides of roadway.

## 6. Conclusions

(a)  The superposition effect of blasting dynamic load between adjacent blastholes in the transmission process leads to the "X" type penetration evolution of surrounding rock fracture area and fracture area between blastholes. Reasonable blast hole diameter and hole spacing can enhance the blasting effect of crack penetration between holes,

and high confining pressure and high rock mass strength can inhibit the blasting effect and the impact effect is small in a certain range.

(b) The design of the blasting roof cutting scheme should focus on the key factors such as hole diameter and hole spacing. The larger the hole diameter, the stronger the superposition effect of blasting dynamic load, and the larger the evolution scale of crack expansion. The hole spacing has a particularly key impact on the effect of roof cutting and crack formation. The confining pressure and rock mass strength have little influence on the expansion and development of blasting cracks, which can be listed as secondary factors for comprehensive analysis in design.

(c) According to the dynamic load transfer of inter-hole blasting, the "X" type penetration evolution characteristics of the broken zone, and broken zone of surrounding rock between holes, the critical criterion of blasting top clearance penetration under the minimum stress environment is deduced. This, combined with the simulation results of four factors affecting the effect of blasting roof cutting, optimizes the design of roof cutting scheme. The feedback of field practice results shows that the cutting scheme design of thick and hard roof has achieved a good blast hole penetration effect.

**Author Contributions:** Conceptualization, M.T. and G.Z.; methodology, X.Z.; software, G.Z. and Q.B.; validation, M.T., Q.B. and J.D.; formal analysis, G.Z. and J.D.; investigation, M.T. and X.Z.; resources, M.T. and X.Z.; data curation, M.T., G.Z. and J.D. All authors have read and agreed to the published version of the manuscript.

**Funding:** This study was funded by the National Natural Science Foundation of China (Nos.52074008, 52074007), the Anhui Collaborative University Innovation Project (GXXT-2020-056).

**Data Availability Statement:** All data generated or analyzed during this study are included in this published article.

**Conflicts of Interest:** The authors declare no conflict of interest.

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
