# Peer review of "Fracture Evolution between Blasting Roof Cutting Holes in a Mining Stress Environment"

_minerals, doi:10.3390/min12040418_

Round 1
Reviewer 1 Report
The paper was well conceived, however the following are to be added to improve the quality of the paper;
1. present a quantitative findings of your results in the abstract. The abstract appear to be more of review paper.
2. References are better cited appropriately in the introduction section
3. Fig. 4, the last digit in the x-axis is not well captured
4. Table 3. The sub-headings are better with their respective units attached e.g. blast hole diameter (mm); Hole spacing (m) etc. Similar trend is to be used for Tables, 5, 6 and 7 respectively.
Reviewer 2 Report
The manuscript describes a study on fracture penetration between blasting holes used for strata preconditioning in underground mines. Through numerical modelling study, the authors obtained the patterns of fracture penetration and the key influencing factors. They also proposed an analytical method to guide the design of the blasting holes. A field application showcases the applicability of the analytical method and the effectiveness of the blasting holes. The manuscript is generally well written and is clear and concise. It can be published with minor revisions.
The following minor comments may help the authors further improve the manuscript.
- The sentence “the diameter of the fracture area is 5–7 times that of the blast hole diameter, and the diameter of the fracture area is approximately 5 times that of the fracture area”, included in both the Abstract and Conclusions are not clear. One of “the diameter of the fracture area” should be “the diameter of the broken/crush zone”.
- The symbols listed in Table 1, e.g. A1, B1, should be described to help readers understand what they denote.
- How was the damage shown in Figure 3 calculated?
- Section 3.2.1 discusses the influence of borehole diameter. Using “aperture” to represent borehole diameter can be confusing and is inconsistent throughout the paper. The last sentence of the first paragraph in this section “the pores can evolve and penetrate” should be changed to “the cracks can evolve and penetrate”.
- There is a mixed use of terms “broken zone” “crush zone”, “fractured area” “fissured zone” throughout the paper and in the figures as well. In addition, the authors should provide a clear definition of these zones.
- There are two symbols of the radius of crush zone: a and r1. ‘a’ might be the refined radius of the crush zone?
- Are equations (1) and (2) derived by the authors. If not, please add the references.
- This sentence is not clear: “The influence of advanced mining is more than 180 m, 80 m in front of the work is affected by mining, and the severe influence range is 60 m in front of the work.”
- In Figure 15 and also in the relevant text, it is more appropriate to use “convergence” instead of “approach amount of roadway” to describe roadway deformation.
Reviewer 3 Report
- The paper needs a general review of English. There are continuous repetitions, starting from the abstract: see the repetitions of the expression " in a mining stress environment"
- Figure 1 is unreadable, please expand
- Section 3: the authors refer to "rock mass", but the medium simulated via software appears to be a continuous, linear, isotropic and elastic one, which a rock mass is not. Discontinuities and heterogeneity are not taken into account. Maybe the authors would like to change "rock"instead of "rock mass"?
- Table 4: "aperture"?? Do the authors mean "diameter"?
- Section 3.2.1 is written in a confused way and results hard to read, it should be re-written in a clearer style
- Section 3.2.1: measures are given in a confused way. Some are in mm, others in cm, it confuses the reader.
- Section 3.2.1: "the larger the aperture, the more explosives are required in the blast hole, and the higher is the blasting dynamic load generated after blasting, which in turn will form a larger rupture gap evolution scale. Only when sufficient detonation energy is achieved can the blast fracture between holes be formed, and the pores can evolve and penetrate."this is merely stating the obvious: larger explosive load causes better fracturing. There is no novelty in this obvious statement, and the whole simulation, as interpreted, provides no novelty for scientific knowledge
- Section 3.2.2: "The analysis shows that the larger the hole spacing, the greater is the blasting dynamic load transfer and attenuation stroke, and the greater is the difficulty in forming the crack penetration in the center of the hole spacing. In addition, even
a sufficiently large distance between the holes cannot realize the effective blast ing and perforation crack evolution.": this again is stating the obvious: larger spacing, lesser hole-to-hole fracturing. No need for a simulation to know this. The simulation, as interpreted, is meaningless - Equations 1 and 2: it is unclear whether they come from literature or are originated by the authors. If the first case, then references should be cited. If the second case, then the authors should show the step-to-step process to come to this formulae.
- Figure 15 is unreadable, please expand
- It is unclear how the formulations of section 4 contribute and/or are used in the experimental work
- Conclusions section (1): ". During blasting top cutting, the broken area of the surrounding rock between holes and the fracture area show “X” type penetration evolution. ": actually this comes from simulation, no field observation of this is shown in the paper
- Conclusions section (2): the authors themselves recognize the content to be "most obvious". The work referred to in this section, and this section itself, is useless as it does not contribute with anything new
- Conclusions section (3): it results confused. Not clear what the authors mean. When fractures between hole meet the roof can be cut off? If so, again this states the obvious
Round 2
Reviewer 1 Report
In the introduction, Jun M J [4], was cited, it is better having it as Jun [4]. How authors need to check through the entire paper and correct this throughout the entire work.
Fig 5, should be well captioned, similar to Fig 2 (better make the caption a, b and define each in the caption).
Similar comments for Figs 7, 8, 9, 12, 13 and 14 respectively
Reviewer 3 Report
Some changes have been made, but the main problem persists: it is a lot of simulation and formulation to come to state the obvious. The bottom line is that the paper offers no scientific/engineering novelty.
